# A Study on the Changes in the Heat Source Capacity and Air-Conditioning Load due to Retrofit; Focusing on a Large Office Building in Korea

**Hyemi Kim [1], Kyung-soon Park [2] , Hwan-yong Kim [3] and Young-hak Song [3],***

[1]   Graduate School of Human-Environment Studies, Kyushu University, Fukuoka 819-0395, Japan; kim@energy.arch.kyushu-u.ac.jp
[2]   Architecture and Civil Engineering, Dong-eui University, Busan 47340, Korea; pks2180@deu.ac.kr
[3]   Department of Architectural Engineering, ERI, Gyeongsang National University, Jinju 52828, Korea; hykim@gnu.ac.kr
*   Correspondence: songyh@gnu.ac.kr; Tel.: +82-55-772-1756

**Abstract:** In an office building, the internal heat and the skin load are both expected to change. Thus, this study is an initial step that searches for optimum replacement measures when a heat source system is replaced on an office building in Korea. The operation status of office buildings in Korea was investigated, and the heat source capacities at the retrofit and the design times were quantitatively studied to determine the optimum capacity during of a heat source during retrofit. For the four retrofit scenarios, the maximum cooling and heating loads were calculated to show that, when retrofit is performed, the maximum cooling and heating loads are decreased by 40%, while the heat source capacity is reduced by around 43%. This is believed to be because of the replacement of the window chassis, which are better sealed with higher heat insulation performance, due to the enhanced design criteria for exterior load designing, and an improved installation process. Concerning the air-conditioning load, the influence of the internal heat load turned out to be significant, indicating that such a factor should be considered when retrofit. Thus, if the heat source capacity at the initial design time is applied equivalently during the office building retrofit, it would lead to degradation in energy efficiency due to the excessive design. Thus, it is necessary to calculate a heat source capacity by reflecting the operational and current status of the load in an actual building at the time of a retrofit.

**Keywords:** retrofit; replacing of source capacity; office building; changing patterns of load

## 1. Introduction

Research on green retrofit has been widely conducted to reduce the energy use in existing buildings around the world [1]. Accordingly, the policies of green retrofit promotion have been executed in Korea to reduce unnecessary energy use in existing buildings according to the energy reduction policies based on the Framework Act on Low Carbon and Green Growth [2]. The study result on retrofits in Korea exhibited that the retrofits ratio was approximately 20% from 2002 to 2012 [3], and residential buildings was 99.3% from 2014 to 2016 [4]. Therefore, in order to reduce the nation-wide energy consumption and carbon emissions, it is necessary to expand green retrofit from the current market of residential buildings to non-residential buildings as well, with more studies and R&D efforts focusing on the issues concerning this goal. Office buildings, which account for a large portion of non-residential buildings are subject to the internal heat loads caused by OA (Office Automation) equipment and occupants, which influence the air-conditioning loads more profoundly when compared to residential buildings. As for their levels, the load from lightings is

around 15%, while OA equipment accounts for around 20% [5]. However, as the service life of lighting and OA equipment ranges from three to five years, they are replaced frequently, which contributed to changes in the internal heat load. A preceding study [6] showed the quantitative picture of such changes.

Meanwhile, the age requirement for buildings to be retrofitted was reduced from 20 years after approval to 15 years, in order to realize the increase of green growth in cities. And usually, the service life of the heat source equipment is expected around 15 years [7,8], which necessitates consideration of replacing the heat source system. However, in previous retrofit methods, the heat source systems were mainly replaced with equipment of the same capacity, and a higher level of efficiency, without adjusting the capacity to reflect changes in loads. If a capacity review is not conducted, this can lead to excessive capacity being applied, resulting in possible partial load operations, or energy consumption increases. Therefore, when retrofit, the assessment of the optimal replacement of the heat source system can be an important factor to consider.

In addition, a study [9] stressed the importance of investigating the maximum cooling load at the time of design since the investigation on the device capacity using the maximum cooling load influenced the system installation cost and device performance. The comprehensive building model and reliable input data are needed to determine the building's maximum load, and most studies have focused on constructing a comprehensive building model. Other studies did not focus on constructing a building's model but on the improvement of uncertain input data. In this regard, they studied the effect of the uncertain input data on the simulated maximum cooling load and searched for a method that could be applied to conservative design. These studies needed the verification of a complex simulation process and were limited to quantifying the practical level of input data with respect to buildings of the same category. In addition, the design criteria in Korea showed a difference when compared to the actual input levels [6,10], which required input data that was simple to apply and reflected the actual operation status of real buildings, and the maximum load, considering that the input data needed to be calculated.

Thus, this research was an initial study to derive the appropriate heat source retrofit for a given office building, in which the heat source capacity was newly derived after identifying the seasonal changes in the internal heat load during practical operation status, and the optimal heat source capacity of the retrofit scenarios was calculated through quantitative comparisons with the heat source capacity at the time of the first design. Through the results of this study, the changing pattern in the maximum cooling load at the time of initial design and by building's life cycle will be identified without going through complex compensation steps. In addition, since the changing pattern of the loads for office buildings whose construction times were different can be identified, the variation of heat source system capacity and the time elapsed after building completion and initial design is expected to be determined. For example, buildings whose completion dates were 1995 and 2005 may have different internal and skin load levels and their changed loads based on the current time can be predicted quantitatively. Accordingly, the change rate in heat source system will also be determined.

## 2. Study Overview

### 2.1. Study Method and Scope

This study is an initial step to the research the optimum replacement measures at the time of a retrofit of heat source systems in an office building. Figure 1 shows the overall research flow, and this study belongs to STEP 2. This study aims to compare building loads quantitatively at the time of design and the time of retrofit and to calculate the optimal heat source capacity at the time of the retrofit. First, the area and scale of the office building, which is represented by one of the buildings in the target area, was selected, and literature reviews were conducted based on the current status considering load factors. Based on the literature review results, the building load was calculated using the RTS-SAREK (Radiant Time Series—The Society of Air-conditioning and Refrigerating Engineers of

Korea program, Seoul, Korea) which was generally used for designs in the target region of this study. In this chapter, a detailed explanation concerning the application of the RTS-SAREK program and the scope of the study will be presented.

### 2.1.1. Literature Reviews

The literature reviews concerning changes in building's internal loads, which correspond to STEP 1 in Figure 1, were conducted in a preceding study [6]. In this study, input values were applied to RTS-SAREK along with the study results by [6], as shown in Table 1. The design values reflected at the time of design, the actual measurement values, and the catalog values for equipment are noted items. The calculated result values were regarded to reflect the current status of the building operation and loads were calculated by equations according to the input values, which is the reason that the correction step was not applied.

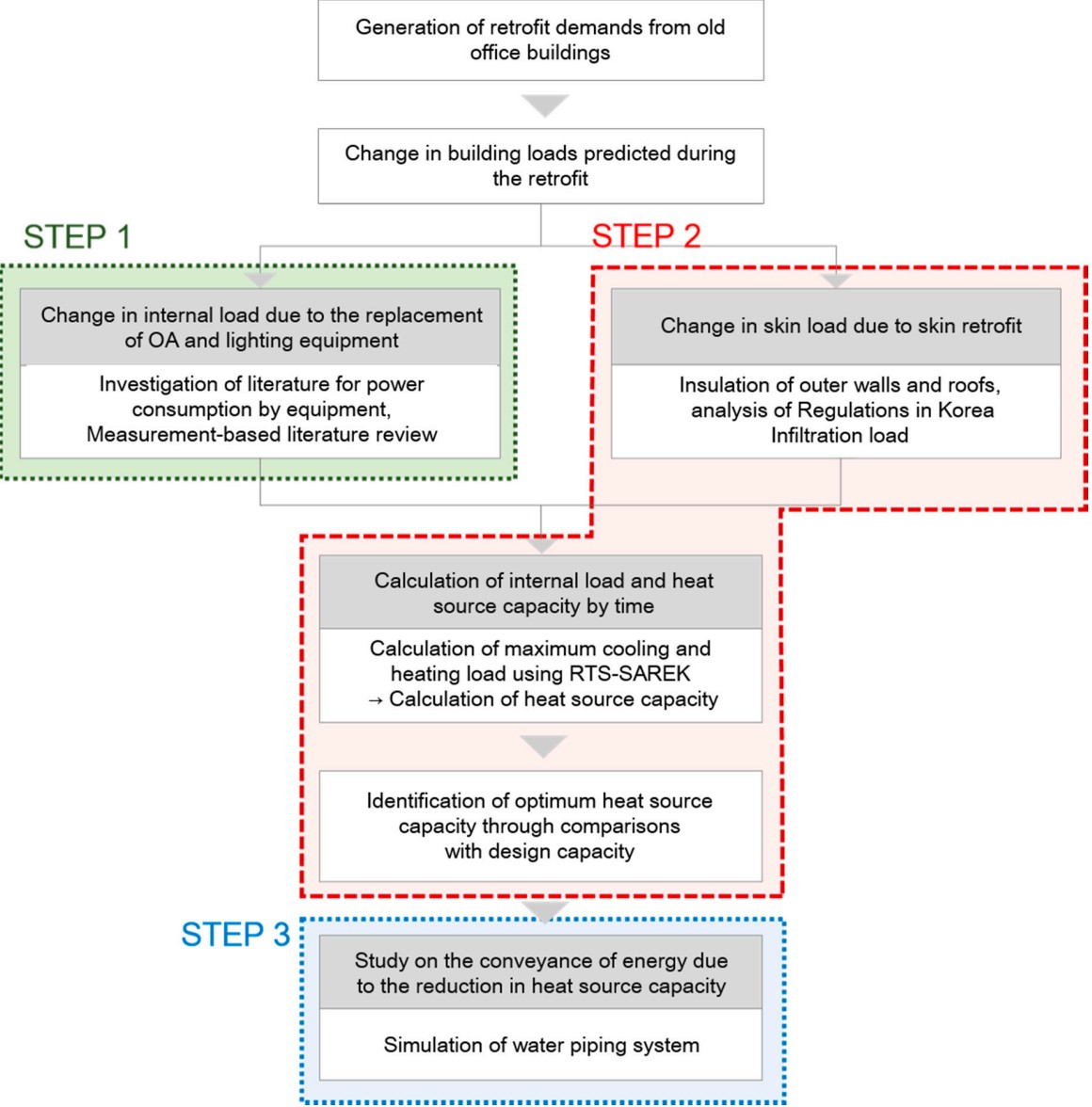

**Figure 1.** Overall research flow chart for the optimal heat source system retrofit.

<div align="center">**Table 1.** Detailed items of literature reviews by load factor.</div>

| Category | | Load Factor | Item |
|---|---|---|---|
| Internal | OA equipment | Desktop | · Electric Power |
| | | Laptop | · Load |
| | | Monitor | · Number of used |
| | | Small-sized printers | machines per person |
| | | All-in-Ones | |
| | Lighting | Fluorescent lamps | |
| | | LED (Lighting-Emitting Diode) | |
| | Occupancy density | | · Load |
| Envelope insulation performance | Exterior wall of the living room | · Enforcement Rules of Construction Act | |
| | Roof on the top floor | · Rules on Equipment in Buildings, etc. | |
| | Bottom of the lowest floor (without floor-heating) | · Energy saving criteria for buildings | |
| | Windows and doors | | |

### 2.1.2. RTS-SAREK

For current the calculation of the cooling and heating loads in buildings, two methods are widely used: the annual load calculation method, which analyzes the total energy and the maximum load calculation method, which is used to select the optimum equipment capacity during building design. Previously, "LOADSYS program (Dangjin, Chungcheongnam-do, Janghan Technology Manufacturing Co., Ltd., Korea)", which were developed in Korea based on the cooling load temperature difference (CLTD)/cooling load factor (CLF) method were used to calculate the building's maximum load in the building design practice, and "Mirae98" based on the CLTD/solar cooling load factor (SCL)/CLF method were developed and applied. These programs were arbitrarily selected and employed to fit the designer's circumstance. However, these programs did not have objective verification processes and their uses are limited due to many factors [11]. To overcome these limitations, the Society of Air-conditioning and Refrigerating Engineers of Korea (SAREK) established a special committee to develop a standardization program for air-conditioning load calculation and developed the maximum cooling load calculation program (RTS-SAREK). Since Version 1.0 of RTS-SAREK was launched in 2006, it has been continuously improved version 5.2 was the most up to date as it was introduced in June 2018.

The program is a certified practical program that makes equipment selection easy based on the radiant time series (RTS) method proposed in the ASHRAE Fundamental, which has been utilized as a standard load calculation program in Korea and it has been heavily utilized in building equipment designer and education purposes. Table 2 presents the yearly spread of RTS-SAREK up to 2013. As of 2018, RTS-SAREK should be considered widely penetrated for use by on-site designers and for educational purposes [12]. Thus, this study employed RTS-SAREK.

<div align="center">**Table 2.** Status of the yearly spread of RTS-SAREK.</div>

| Category | 2006 | 2007 | 2008 | 2009 | 2010 | 2011 | 2012 | 2013 | Total |
|---|---|---|---|---|---|---|---|---|---|
| Education | 2 | 62 | 105 | 40 | 38 | 65 | 169 | 1 | 482 |
| General | 25 | 101 | 55 | 66 | 78 | 82 | 81 | 9 | 497 |
| Total | 27 | 163 | 160 | 106 | 116 | 147 | 250 | 10 | 979 |

The cooling load calculation method of RTS-SAREK complied with the computation method of the RTS method, which was a cooling and heating load calculation method published in Chapter 18 of the ASHRAE Fundamental in 2009. The application scope and calculation range of the program are presented in Table 3.

**Table 3.** Application scope and calculation range of RTS-SAREK [12].

| Category | Limitation and Items | Category | Limitation and Items |
|---|---|---|---|
| Solar radiation-related calculation<br>- Solar heat gain<br>-Solar Air Temperature | All regions in the northern and southern hemispheres | Internal heat gain | Human body heat load (sensible and latent heat), lights, and appliance loads |
| Design outdoor air temperature and humidity | Cooling: 50 °C or lower Heating: −50°C or higher | Infiltration of outdoor air-cooling load | Air change/direct input |
| K value calculation<br>- Glass<br>- Wl/Rf/Pt | Glass: 15 items R/W/P: 40 items | Total cooling load | Sum of radiation/convection load, glass's conduction load, partition load, and infiltration outdoor air load |
| CTS coefficient | Wall/Roof: 35/19 items | Heating load | External (G/R/W), partition load and glass load |
| RTS coefficient | Zone: 24 items | No. of rooms | 1600 rooms |
| Glass heat gain<br>- SHGC value<br>- IAC value | Direct/Diffuse/ Conduction heat gain | No. of AHU and terminal unit rooms | 190 items/system |
| Wall/Roof heat gain | Hourly solar air temperature difference/ CTS coefficient | Equipment capacity-selectable system types | - CAV AHU<br>- VAV AHU<br>- FCU<br>- FCU + CAV AHU<br>- FCU + VAV AHU<br>- CAV AHU with Reheat<br>- OAHU or HVU<br>- PAC<br>- Radiator/Convector<br>- Heat Recovery |

Moreover, a study by [13] conducted a load analysis of HVAC (Heating, Ventilation, and Air Conditioning) systems designed using RTS-SAREK, and his study results showed that RTS-SAREK could select similar air-conditioning equipment as compared to other load calculation programs, resulting in a high utilization in practice. In addition, RTS-SAREK can be utilized immediately for the follow-up work and it promotes user-friendliness, efficiency, promptness, and accuracy by shortening the time taken for calculations by relieving cumbersome tasks, such as the repeated input of reference data and use purpose/room name/systems name by simply coding all utilized data.

Thus, this study is conducted by applying RTS-SAREK generally used in the design offices in Korea as this study target office building. RTS-SAREK calculates the maximum cooling load, as shown in the following Figure 2. This study aims to identify the changing pattern of loads according to the conditions by using thermal transmittance and room data. Through this, the peak load of the building is calculated, and the capacity of the heat source equipment is calculated.

The study results concerning the internal heat load were applied from the results of the previous study [6], and the thermal transmittance, required to calculate the skin load, was applied by studying the design criteria such as the Building Acts in Korea at the time of construction. Also, based on the measurements in [14], the infiltration value per unit area of the window was applied. In the case of the maximum heating load, the same calculation method as with the cooling load was used, except that the load was calculated without including the internal heat load or input from sunlight.

### 2.1.3. Heat Source Capacity Calculation Method

Using the maximum cooling and heating load calculated by RTS-SAREK, the heat source capacity for cooling and heating were calculated. The air-conditioning heat source capacity was calculated using the building load plus heat loss occurring due to piping, pumping load, and equipment heat accumulation, while the heating heat source capacity was calculated using the building load plus the heated water load, considering piping heat loss, and warming loss. According to [15–17], the heat source capacity increased by approximately 4% to 15% when compared to the designed building load. Therefore, in this study, an additional margin of 10% was added, leading to the calculation of the heat source capacity increasing by 20%, accounting for both the maximum cooling and heating loads.

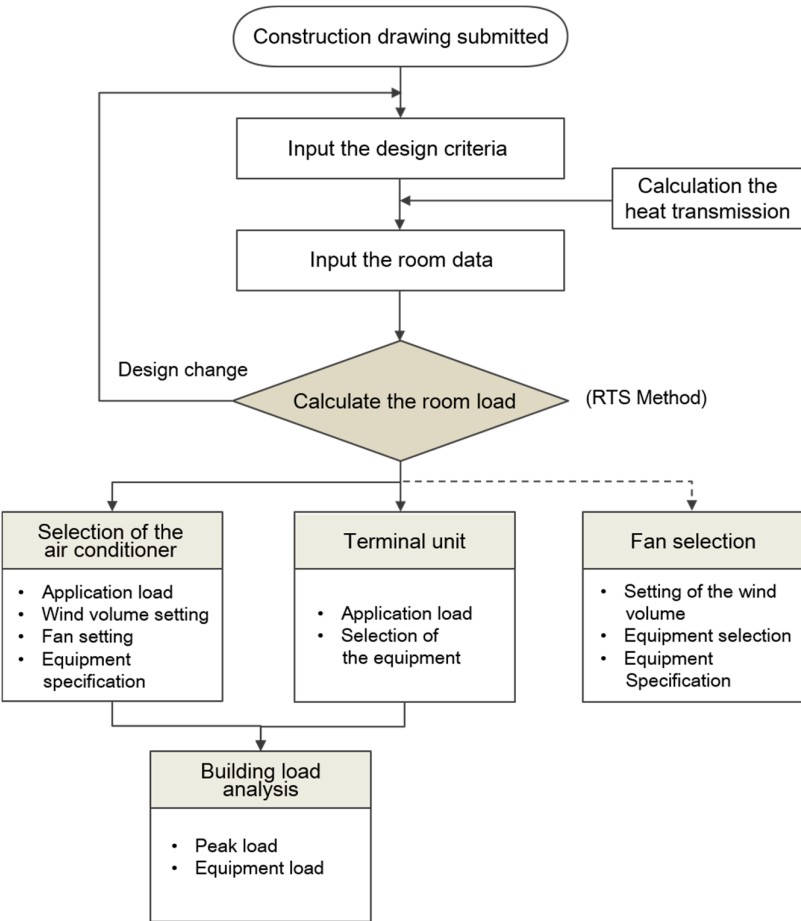

**Figure 2.** Load calculation process of RTS-SAREK program.

### 2.2. Scope of Study

Korean law states that old buildings in poor conditions, unless they are not apartment or joint residential buildings, need to be at least 30 years old, however, the age requirement for buildings is less demanding in the case of retrofit, which is only 15 years. Therefore, the entire study period was set as 30 years, from 1985 to 2015, and retrofit was assumed to have taken place after 15 years. The detailed retrofit scenarios during the study period are as follows;

### 2.2.1. Overview of the Target Office Building

The target region in this study was Seoul Korea, which is characterized by cold dry-winter and hot and humid summer resulting in a classification by the Köppen climate classification system. Considering this climate, the design conditions were applied and the representative office size was investigated to select the target building.

In order to select the target office building for this study, the average gross floor area of office buildings, which was calculated by [18] to be 31,770 m², and the air-conditioning area ratio of 63%, which was calculated by [19] were used. The window area ratio which determines the amount of infiltration and sunlight was set to be around 50%. As for the detailed size of the building, a large office building in Energy Plus was used, having a width of 73 m, and a depth of 48 m, making the area of the reference floor area 3432 m². There was a total of nine floors, with the gross floor area of 30,886 m², while the air-conditioned area was 20,613 m². The exterior and interior of the reference floor was designed to have 10 zones, as shown in Figure 3, leading to a total to 90 zones in the entire building.

The office hours of the building were from 8:00 AM until 8:00 PM. The detailed design conditions, in which the values used in energy saving designs [20], are shown in Table 4. In consideration of the scope of the study, the author selected the fixed wind volume AHUs and FCUs, which were commonly used during 1980s, while the specification of such equipment was in accordance with the default values provided by RTS-SAREK.

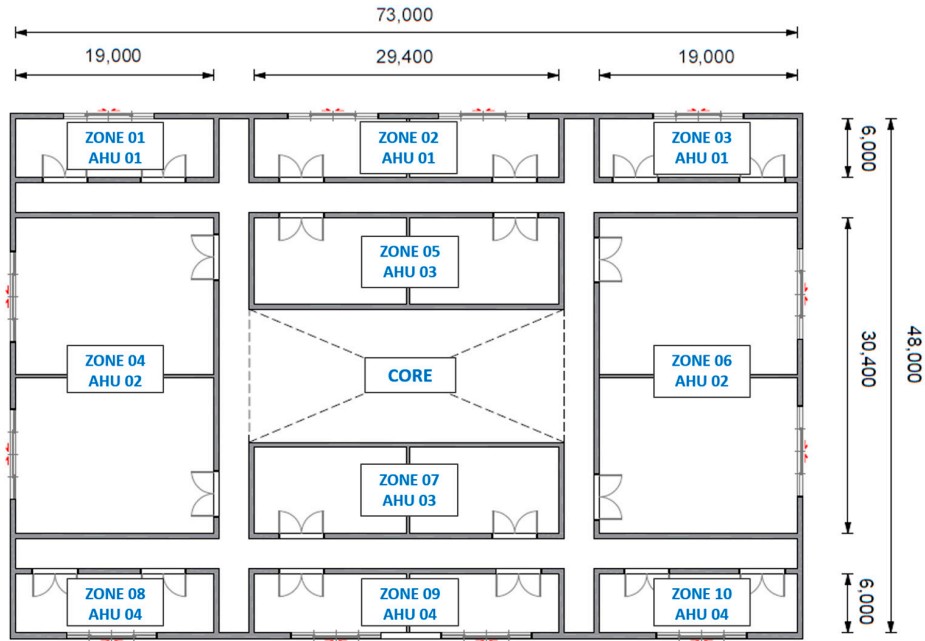

**Figure 3.** Zoning of the target office reference floor (Unit: mm).

**Table 4.** Detailed design conditions for construction and heat source equipment.

| Construction Design Condition | | | Heat Source Equipment Design Conditions | | |
|---|---|---|---|---|---|
| Location | Seoul | | System | FCU + CAV AHU type | |
| Interior air-conditioning conditions | Dry bulb temperature (°C) | 26 | Air-conditioning coil water temperature | Entrance temperature (°C) | 7 |
| | Relative humidity (%) | 50 | | Exit temperature (°C) | 12 |
| Interior heating conditions | Dry bulb temperature (°C) | 20 | Heating coil water temperature | Entrance temperature (°C) | 80 |
| | Relative humidity (%) | 40 | | Exit temperature (°C) | 70 |
| Design for summer atmospheric conditions | Dry bulb temperature (°C) | 31.2 | Conditions of the discharged air | Dry bulb temperature (°C) | 18 |
| | Relative humidity (%) | 63.6 | | Relative humidity (%) | 53~56 |
| | Daily range (°C) | 10 | | | |
| Winter-season design atmospheric conditions | Dry bulb temperature (°C) | −11.3 | FCU air-conditioning load | Sensible heat (W) | 2354 |
| | Relative humidity (%) | 63 | | Latent heat (W) | 2500 |
| Temperature gap with a non-airconditioned room (°C) | 2 (same for cooling and heating) | | FCU heating load | 4000 (W) | |

Figure 4 shows the heat source system diagram for the target building. In this study, the absorption type water cooling and heating system were selected, as they can address both cooling and heating applications. Two absorption type water cooling and heating systems were used for multiple unit application of partial loads. A commercial model provided by S company was used to determine the equipment capacity at the time of design and the time of retrofit and these values were compared.

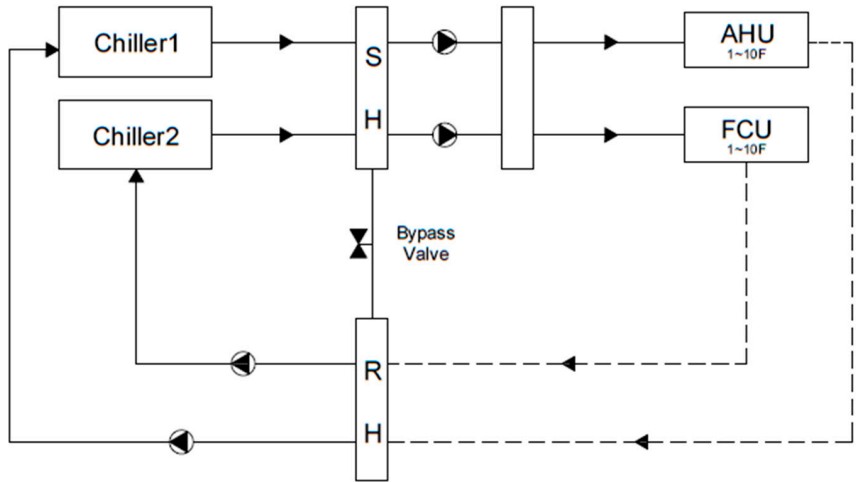

**Figure 4.** Heat source system diagram for the target office.

### 2.2.2. Overview of the Scenarios

This study calculated the results for each of the four scenarios by using the same building information. The selection order of the four scenarios are as follows, and the detailed contents for each scenario are shown in the corresponding figure. Scenario I was created first followed Scenarios II, III, and IV, which were created to determine changes in the building's maximum due to design time. The composition of Scenario I is as follows:

(1) It is based on large office buildings completed more than 30 years ago (meeting the definition of an old building).

(2) Assumption that a retrofit is needed every 15 years. Thus, first and second retrofits occur in 2000 and 2015 respectively, assuming the building was built in approximately 1985.

(3) The loads are calculated every five years to identify the status of changes in buildings loads and due to the life expectancy of internal heat appliances.

(4) Once the item levels at the design and retrofit times are determined, the levels for each item following the design time and preceding the retrofit time are considered (Table 5)

The detailed setup values for the item level for Scenario I, are presented in the following Table 5. The year represents which input values were considered to calculate loads according to the design time, prior to the retrofits. For internal heat equipment, five-year interval values were applied. For occupancy density, the same level was maintained. For envelope insulation performance, the level of design was maintained before a retrofit. For infiltration, aging was applied.

**Table 5.** Items in Scenario I and input values applied by time.

| Year | Internal Heat of Equipment | Occupancy Density | Envelope Insulation Performance | Infiltration |
|------|---------------------------|-------------------|--------------------------------|--------------|
| 1985 | 1985 | 1985 | 1985 | 1985 |
| 1990 | 1990 | 1985 | 1985 | 1990 |
| 1995 | 1995 | 1985 | 1985 | 1995 |
| 2000 | 2000 | 1985 | 2000 | 2000 |
| 2005 | 2005 | 1985 | 2000 | 2005 |
| 2010 | 2010 | 1985 | 2000 | 2010 |
| 2015 | 2015 | 1985 | 2015 | 2015 |

In the four scenarios (Figure 5), the maximum loads at the design and the retrofit times can be compared quantitatively, and the status of loads in large office buildings by design time can be determined through the comparison between the scenarios. In addition, the status of changes in loads due to retrofit can be identified in the scenarios and the operational status of the actual office buildings can be affected by legal regulation strengthening and technical advancement. Thus, this study configured four scenarios with five-year intervals.

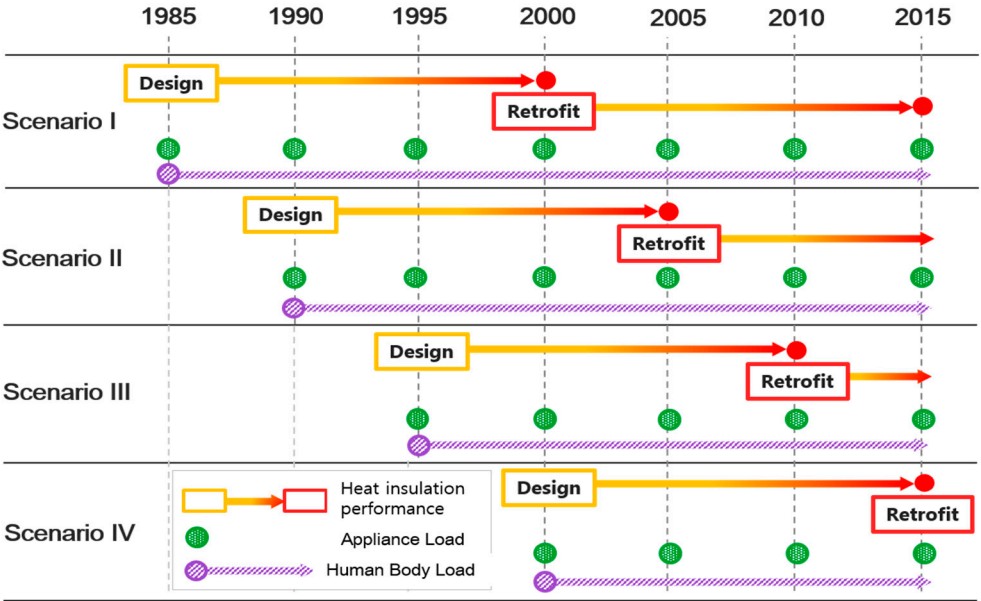

**Figure 5.** Retrofit and scenario overview.

## 3. Results of Calculation of the Building Load and Heat Source Capacity

Before calculating the maximum cooling and heating load, the interior, skin loads, and infiltration were examined via a literature review, in order to reflect the actual conditions in South Korea. These values were calculated in line with the conditions of the target building of this study. The calculated values for different load items were applied to RTS-SAREK with regard to different time periods, for calculating the maximum cooling load, the heating load, and the heat source capacity. The details of these processes are as shown below.

### 3.1. Internal and Skin Load Survey Results

#### 3.1.1. Result of the Survey on the Internal Heat Load

The internal heat sources were OA equipment, lighting, and human body heat. In a preceding study [6], the loads were calculated based on literature and catalogues. Table 6 shows the result of a survey on electricity consumption per unit of OA equipment and lighting. The numbers of items did not match between the time periods, so the weighted average of each five-year-interval was used.

**Table 6.** Weighted average of the power consumption for each internal heat source (Unit: W).

| Year | Desktop | Laptop | Monitor | Small-Sized Printers | All-in-Ones | Fluorescent Lamps | LED |
|------|---------|--------|---------|----------------------|-------------|-------------------|-----|
| 1985~1989 | 178.9 | 16.9 | 40.6 | 39.6 | N/A | N/A | N/A |
| 1990~1994 | 244.0 | 39.9 | 59.2 | 325.7 | N/A | 27.4 | N/A |
| 1995~1999 | 205.4 | 47.9 | 88.9 | 155.2 | N/A | 29.3 | N/A |
| 2000~2004 | 305.5 | 47.8 | 64.6 | 401.4 | N/A | 31.3 | N/A |
| 2005~2009 | 152.8 | 80.9 | 49.3 | 346.5 | 1323.9 | 30.4 | 7.3 |
| 2010~2014 | 120.4 | 45.0 | 36.0 | 450.5 | 1175.8 | 31.9 | 18.3 |
| 2015~ | 53.3 | 42.7 | 32.0 | 495.0 | 1301.8 | 32.6 | 17.2 |

The results of the survey showed that the power consumption for desktop computers and laptops increased until the early 2000s and then started to take a downturn. This appears to be due to the demands of the customers and development of stand-by power consumption reduction technology to enhance the energy efficiency after the development of high-performance computers. In the case of printers, the power consumption was on the rise due to the penetration of laser-type printers and the increased speed of printing. Since 2005, the internal heat generation also increased due to introduction of all-in-one appliances. As for lighting, LED lamps started to be used in 2005, and were found to have been used in retrofit projects as well as in new construction projects since 2010.

Table 7 shows the number of OA appliances used per person, and the density of occupants. Since 2005, office workers were found to use one computer per person, while the office occupant density showed virtually no change.

**Table 7.** The number of OA appliances used per person and the density of occupants.

| Year | Computer (Unit/person) | Monitor (Unit/person) | Small-Sized Printers (Unit/person) | All-in-Ones (Unit/person) | Occupant Density (Person/m$^2$) |
|---|---|---|---|---|---|
| 1985~1989 | 0.7 | 0.85 | 0.25 | N/A | 0.083 |
| 1990~1994 | 0.7 | 0.85 | 0.25 | N/A | 0.108 |
| 1995~1999 | 0.75 | 0.85 | 0.263 | N/A | 0.073 |
| 2000~2004 | 0.81 | 0.86 | 0.343 | N/A | 0.069 |
| 2005~2009 | 1 | 1 | 0.1 | 0.025 | 0.072 |
| 2010~2014 | 1 | 1 | 0.1 | 0.025 | 0.095 |
| 2015~ | 1 | 1 | 0.1 | 0.025 | 0.090 |

Table 8 shows the internal heat load per unit area, calculated using the data in Tables 6 and 7. The printer load is entirely reflected in the number of units used per person for small-sized printers and all-in-ones. The load from OA equipment is the sum of the load of desktops, monitors, and printers. Also, the lighting load is based on a 2010 survey, conducted by Korea Energy Corporation, which averaged the load in 24 private office buildings, along with the average LED application ratio in 2806 non-residential buildings, which was between 67 and 77%.

**Table 8.** Calculation result for internal heat load per unit area by items.

| Year | OA Equipment (W/m$^2$) | | | | | Fluorescent Lamps (W/m$^2$) | LED (W/m$^2$) | Human Body Emission (W/m$^2$) |
|---|---|---|---|---|---|---|---|---|
| | Desktop | Laptop | Monitor | Printer | Sum | | | |
| 1985~1989 | 10.37 | 0.98 | 2.86 | 0.82 | 14.05 | N/A | N/A | 9.82 |
| 1990~1994 | 18.39 | 3.01 | 5.42 | 8.77 | 32.58 | 8.71 | N/A | 12.77 |
| 1995~1999 | 11.15 | 2.60 | 5.49 | 2.96 | 19.59 | 9.30 | N/A | 8.61 |
| 2000~2004 | 16.97 | 2.65 | 3.80 | 9.43 | 30.20 | 9.96 | N/A | 8.13 |
| 2005~2009 | 11.00 | 5.83 | 3.55 | 15.04 | 29.59 | 9.66 | 2.32 | 8.54 |
| 2010~2014 | 11.47 | 4.28 | 3.43 | 19.37 | 34.26 | 10.14 | 5.81 | 11.30 |
| 2015~ | 4.79 | 3.84 | 2.88 | 20.21 | 27.89 | 10.37 | 5.46 | 10.67 |
| Average | 12.02 | 3.31 | 3.92 | 10.94 | 26.88 | 9.69 | 4.53 | 9.98 |

The survey on the internal heat load showed that the load from OA equipment increased by 49.6% over the past 30 years. This is believed to be mainly attributable to the introduction of large all-in-one machines, which pushed the load from printers upward. In the meantime, the average load from fluorescent lamps was 9.7 W/m$^2$, while that of an LED was around 4.53 W/m$^2$, resulting in a 47% load reduction. The loads in Table 8 were identified via articles, live measurements, reports, and equipment catalogues, and are believed to be close to the actual internal heat loads in offices of in South Korea for each period.

### 3.1.2. Result of the Skin Load Survey

The heat insulation performances of the exterior walls and windows have been subject to rules that been continuously strengthened. The details of these are as shown in Table 9. In South Korea, the Enforcement Rules of Construction Act stated legal regulations concerning the heat insulation performance of buildings. These criteria were further strengthened when the Rules on Equipment Criteria for Buildings was introduced in 2001. Recently, the government introduced 'Energy Saving Designing Criteria' in 2013, to prevent excessive heat loss in buildings, and to promote green buildings techniques, by strengthening the requirements continuously. As a result, the heat insulation performance of the exterior walls that contact the atmosphere directly was enhanced by 53%, while the roof performance was enhanced by 69%, the floor by 50%, and windows by 40%.

**Table 9.** Changes in the thermal transmittance of envelope design criteria in South Korea (Central region, not including multi-unit houses or apartments) (Unit: $W/m^2 \cdot K$).

| Year | Exterior Wall of The Living Room | | Roof on the Top Floor | | Bottom of the Lowest Floor (Without Floor-Heating) | | Windows and Doors | | Remarks |
|---|---|---|---|---|---|---|---|---|---|
| | | | | | Atmosphere | | | | |
| | Direct | Indirect | Direct | Indirect | Direct | Indirect | Direct | Indirect | |
| 1984 | 0.58 | 0.47 | 0.58 | N/A | 0.58 | N/A | 3.49 | N/A | Enforcement Rules of Construction Act |
| 1987 | 0.58 | 0.47 | 0.41 | N/A | 0.58 | N/A | 3.37 | N/A | |
| 2001 | 0.47 | 0.64 | 0.29 | 0.41 | 0.41 | 0.58 | 3.84 | 5.47 | Rules on Equipment in Buildings, etc. |
| 2008 | 0.47 | 0.64 | 0.29 | 0.41 | 0.41 | 0.58 | 3.4 | 4.6 | |
| 2010 | 0.36 | 0.49 | 0.2 | 0.29 | 0.41 | 0.58 | 2.4 | 3.2 | |
| 2013 | 0.27 | 0.37 | 0.18 | 0.26 | 0.29 | 0.41 | 2.1 | 2.6 | Energy saving criteria for buildings |

### 3.1.3. Determination of Infiltration

The average ventilation in 1985 was set to 2.0 times/h. In Table 10, the value of 57.4 $m^3$/h in Scenario I for 1985 was a conversion of this information into infiltration per unit areas of the size of the office in question for this study. After this, the study results of [21,22], which actually measured the amount of infiltration per unit area, were used to calculate the increase in infiltration on average per year after completion, which was found to be 1.168 $m^3$/h.

**Table 10.** Infiltration in different scenarios (Unit: $m^3$/h).

| Year | Scenario I | Scenario II | Scenario III | Scenario IV |
|---|---|---|---|---|
| 1985 | 57.4 | n/a | n/a | n/a |
| 1990 | 62.0 | 49.9 | n/a | n/a |
| 1995 | 67.9 | 54.6 | 42.5 | n/a |
| 2000 | 35.0 | 60.4 | 47.1 | 35.0 |
| 2005 | 39.7 | 27.4 | 53.0 | 39.7 |
| 2010 | 45.5 | 32.2 | 20.1 | 45.5 |
| 2015 | 14.3 | 38.1 | 24.8 | 14.3 |

Based on the survey and calculation results above, the load values calculated to be applied to RTS-SAREK for the items are as follows (Table 11): As for the skin load, the legal requirements for the time period were applied to RTS-SAREK. The internal heat load was applied by five-year periods. Also, the occupants of the offices were considered to of the same at the time of designing and the time of retrofit. The amount of infiltration was based on the values in Table 10, according to RTS-SAREK.

**Table 11.** The results of calculation of the skin loads and internal loads for each scenario.

| Scenario | Year | Heat Insulation Performance of The Walls and Windows (W/m²·K) | | | | Internal Heat Load | | |
|---|---|---|---|---|---|---|---|---|
| | | Exterior Wall | Partition Wall | Roof | Windows | Appliances (W/m²) | Lighting (W/m²) | Human Body (person/m²) |
| Scenario I | 1985 | 0.58 | 0.64 | 0.58 | 3.5 | 29 | 11 | 0.135 |
| | 1990 | 0.58 | 0.64 | 0.58 | 3.5 | 25 | 9 | 0.135 |
| | 1995 | 0.58 | 0.64 | 0.58 | 3.5 | 15 | 9 | 0.135 |
| | 2000 * | 0.47 | 0.64 | 0.29 | 3.8 | 23 | 10 | 0.135 |
| | 2005 | 0.47 | 0.64 | 0.29 | 3.8 | 27 | 10 | 0.135 |
| | 2010 | 0.47 | 0.64 | 0.29 | 3.8 | 31 | 6 | 0.135 |
| | 2015 * | 0.27 | 0.37 | 0.18 | 2.1 | 27 | 6 | 0.135 |
| Scenario II | 1990 | 0.58 | 0.64 | 0.41 | 3.4 | 29 | 11 | 0.135 |
| | 1995 | 0.58 | 0.64 | 0.41 | 3.4 | 15 | 9 | 0.135 |
| | 2000 | 0.58 | 0.64 | 0.41 | 3.4 | 23 | 10 | 0.135 |
| | 2005 * | 0.47 | 0.64 | 0.29 | 3.8 | 27 | 10 | 0.135 |
| | 2010 | 0.47 | 0.64 | 0.29 | 3.8 | 31 | 6 | 0.135 |
| | 2015 | 0.47 | 0.64 | 0.29 | 3.8 | 27 | 6 | 0.135 |
| Scenario III | 1995 | 0.58 | 0.64 | 0.41 | 3.4 | 18 | 19 | 0.115 |
| | 2000 | 0.58 | 0.64 | 0.41 | 3.4 | 23 | 10 | 0.115 |
| | 2005 | 0.58 | 0.64 | 0.41 | 3.4 | 27 | 10 | 0.115 |
| | 2010 * | 0.36 | 0.49 | 0.20 | 2.4 | 31 | 6 | 0.115 |
| | 2015 | 0.36 | 0.49 | 0.20 | 2.4 | 27 | 6 | 0.115 |
| Scenario IV | 2000 | 0.47 | 0.64 | 0.29 | 3.84 | 50 | 22 | 0.110 |
| | 2005 | 0.47 | 0.64 | 0.29 | 3.84 | 27 | 10 | 0.110 |
| | 2010 | 0.47 | 0.64 | 0.29 | 3.84 | 31 | 6 | 0.110 |
| | 2015 * | 0.27 | 0.37 | 0.18 | 2.10 | 27 | 6 | 0.110 |

* When retrofit was carried out.

*3.2. Calculation Result of the Max Cooling and heating Loads and Reviews*

Table 12 shows the calculation of maximum cooling and heating load for the entire scenario. Scenarios I and IV show the lowest reduction rates, details of which are described below. Figure 6 shows seasonal changes in infiltration load for each scenario, and Figure 7 shows the changes in external loads. Figure 8 also shows the change in internal load according to the scenario. Scenario I will be described as a representative example, and scenarios II to IV will be described focusing on differences from Scenario I to Scenario IV.

**Table 12.** The results of calculating the maximum cooling and heating load for the entire scenario.

| Category | | Scenario I | Scenario II | Scenario III | Scenario IV |
|---|---|---|---|---|---|
| Maximum cooling load (kW) | Initial design | 3134 | 2991 | 2734 | 3322 |
| | Retrofit | 2296 / 2288 | 2691 | 2325 | 2110 |
| | Reduction rate (%) | 27 / 27 | 8 | 15 | 36 |
| Maximum heating load (kW) | Initial design | 2623 | 2455 | 2189 | 2039 |
| | Retrofit | 1823 / 1559 | 2171 | 1591 | 1380 |
| | Reduction rate (%) | 30 / 41 | 11 | 27 | 32 |

3.2.1. Scenario I

In Scenario I, an office building that was designed in 1985 was retrofitted in 2000 and 2015. During the initial retrofit, the cooling load was reduced by around 30%, while the heating load was reduced by around 34%. During the second retrofit in 2015, the cooling load was decreased by 34% as compared

to the initial design, while the heating load was down by 47%. It is believed this is because of an approximate 39% decrease in the cooling and heating infiltration load and a 26% internal heat load reduction during cooling (Figure 9).

Also, compared to the first retrofit, the second retrofit saw a reduction of around 5% in the cooling load, while the heating load decreased around 19%. This is believed to be because of the reduction of the infiltration by approximately 60%, as compared to the first retrofit. The load element that affected both cooling and heating was infiltration, which increased by 20% or more due to corrosion and aging until retrofit. Also, the infiltration was reduced by around 75% compared to the initial designing, due to the use of window chassis with higher tightness and heat insulation performance during the second retrofit.

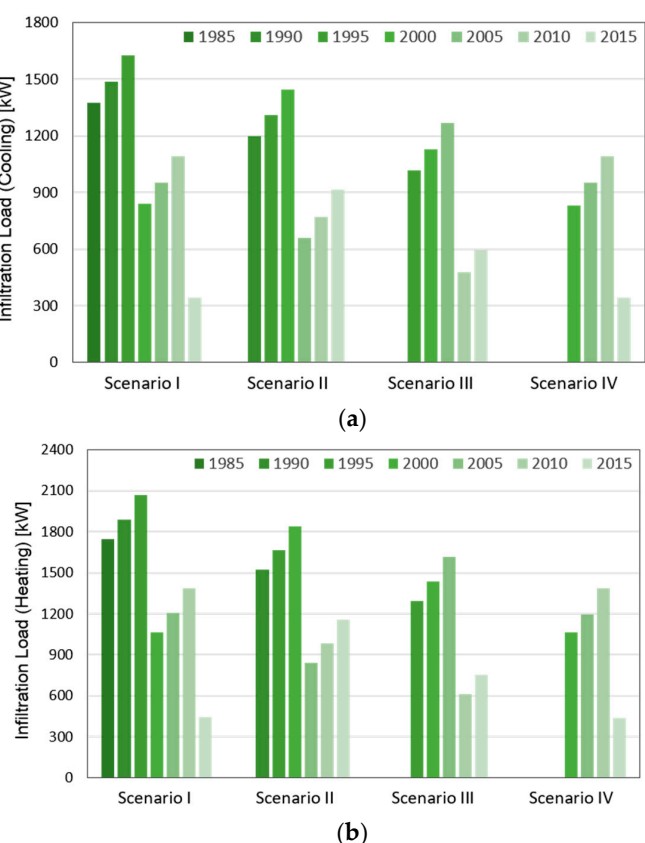

**Figure 6.** (**a**) is changes in the load of infiltration per scenario in cooling period, while (**b**) is the result in heating period.

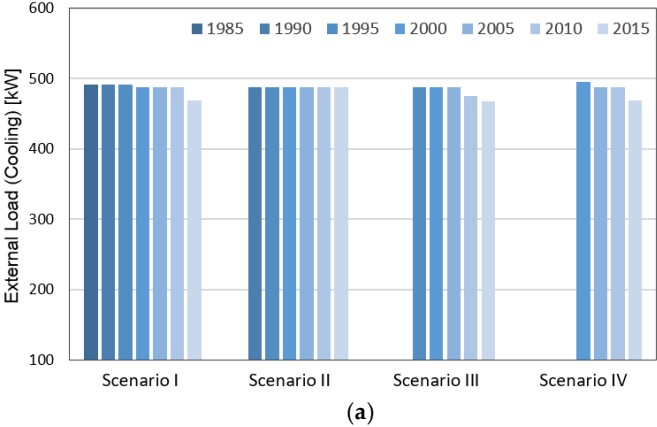

**Figure 7.** *Cont.*

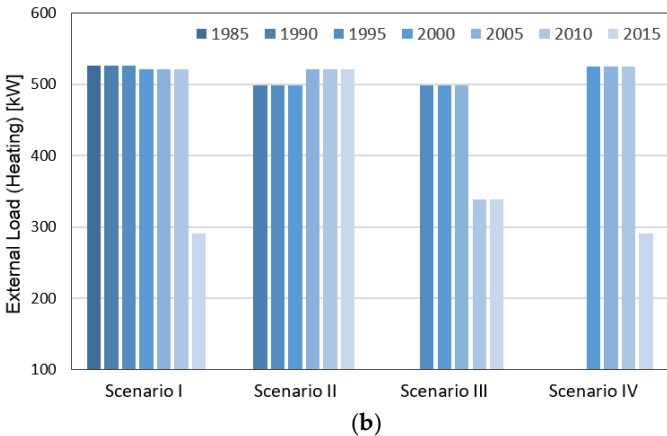

**Figure 7.** (**a**) is changes in the load of external per scenario in cooling period, while (**b**) is the result in heating period.

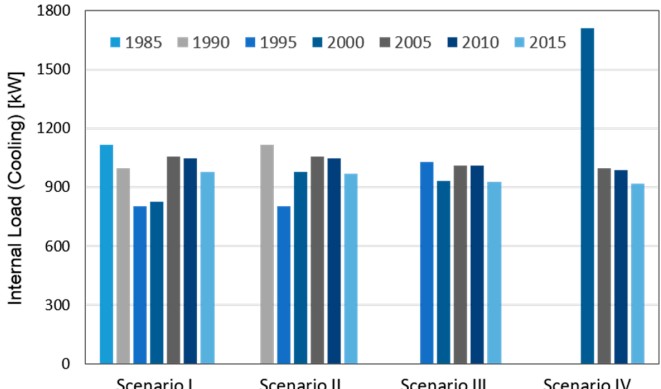

**Figure 8.** The result of the internal load per scenario.

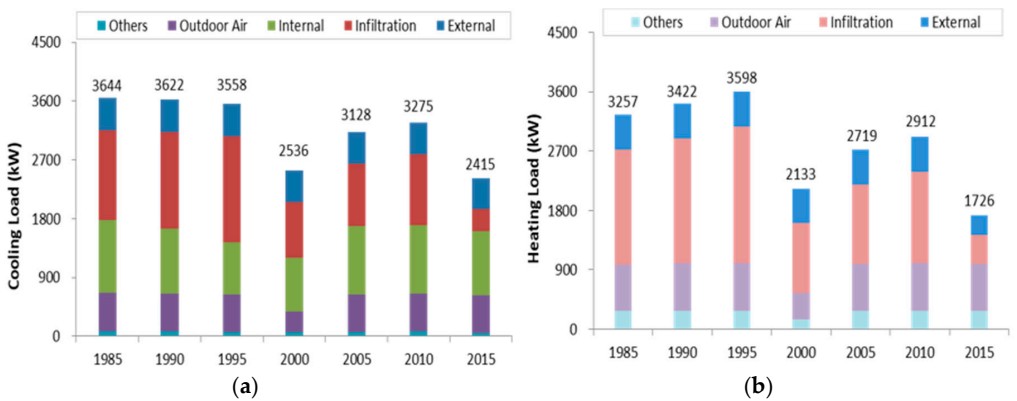

**Figure 9.** (**a**) is the maximum cooling, while (**b**) is the maximum heating load calculation result for Scenario I.

In the case of changes in skin loads, the influence of the first retrofit was negligible, while the second retrofit resulted in a reduction of approximately 5%. This is believed to be because of the strengthening of the design criteria introduced in 2013. In addition, the internal heating load initially fluctuated for 30 years after the initial design period, before it plunged by a maximum of 28%. It is believed that the changes in the load from OA equipment affected the final changes of cooling loads. On the other hand, the 2016 version of the Building Energy Efficiency Grading System shows that the reference OA load is around 50.4 W/m$^2$. This is significantly different from the value identified in this study, which was 33 W/m$^2$. Therefore, it was confirmed that, when designing or retrofit a building, the OA load must be evaluated quantitatively for the target building.

### 3.2.2. Scenario II~IV

Figure 10 shows the max. cooling and heating loads for Scenarios II, III, and IV. In most cases, the changes had similar patterns with those of Scenario I. Here, the influence of the infiltration load was profound. Therefore, the results of Scenarios II, III, and IV will be described focusing on the differences with those of Scenario I.

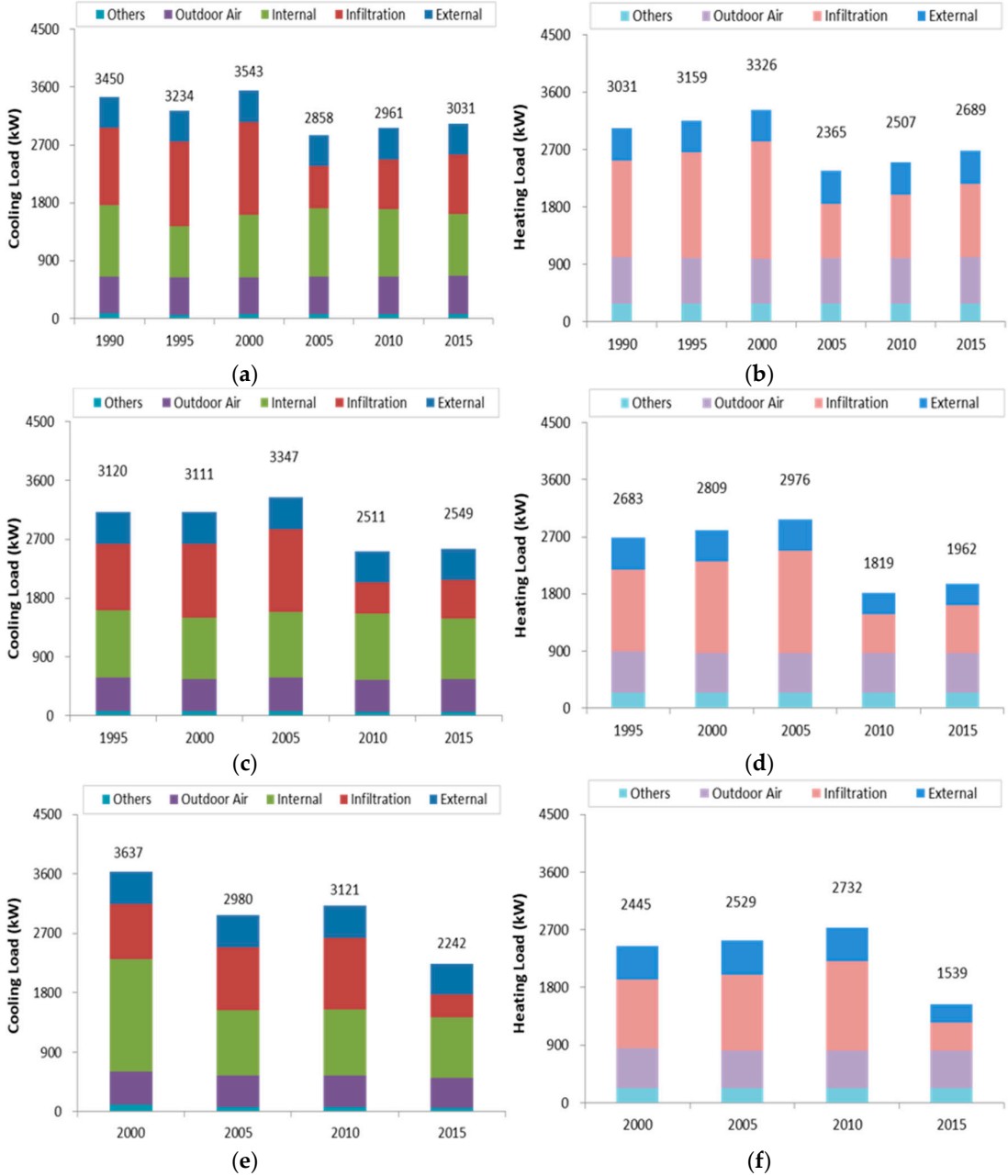

**Figure 10.** The result of the max. cooling and heating loads for Scenario II through IV: (**a**) The result of the max. cooling load calculation for Scenario II; (**b**) Result of the max. heating load calculated for Scenario II; (**c**) Result of the calculation for the max. cooling load for Scenario III; (**d**) The result of the calculation of max. heating load for Scenario III; (**e**) The result of the max. cooling load calculation for Scenario IV;(**f**) Result of the max. heating load calculated for Scenario IV.

Scenario II assumes that the building was designed in 1990 and was retrofitted in 2005. In this scenario, the cooling load was reduced around 17%, while the heating load was reduced by around 20%. The factor that affected the changes in the loads most significantly was the infiltration. The infiltration

load increased by around 21% until retrofit occurred. As for the difference with Scenario I, the max. cooling was reduced compared to the initial design due to the reduction of the internal heating load. This showed that, in large-scale offices, the load from the OA equipment had a significant impact during cooling.

Scenario III assumes that the building was designed in 1995 and was retrofitted in 2010. In this scenario, the cooling load was reduced by approximately 20%, while the heating load was reduced by approximately 32%. Also, in Scenario III, the internal heat load decreased by around 10% after five years from the completion of the design, reducing the max. cooling load accordingly.

In Scenario IV, it was assumed that an office that was designed in 2000 was retrofitted in 2015. In this scenario, the cooling load was reduced by approximately 38%, and the heating load by approximately 37%. In Scenario IV, the internal heating load increased by more than 40% compared to the time of design. Also, compared to the internal heat load in Scenario I, the internal heat load of Scenario IV was 53% higher. It is believed that this is because of the introduction of high performance, high efficiency OA equipment in the 2000s, which increased the cooling load as compared to other scenarios. Also, during retrofit, the reduction of the cooling load was the most significant, which is believed to be because of the introduction of the building energy saving design criteria that boosted the tightness and skin heat insulation performance.

*3.3. Result and Review of the Calculation of the Heat Source Capacity*

In this study, the heat source capacity was calculated using the max. cooling and heating load calculated for each scenario. The results of the calculation of the max. heating and cooling loads using the RTS-SAREK program were increased by another 20% as a buffer to calculate the heat source capacity. With this, the capacity of absorption type water heating/cooling equipment in [23] (pp. 16–19) was determined within the range of the cooling capacity 1407 ~2461 kW and the heating capacity 1267~2215 kW. As the purpose of this study is to review the changes in heat source capacities due to retrofit, the equipment with the same efficiency and specification were used.

Table 13 shows the changes in the heat source capacity due to retrofit as compared to the initial design. As the absorption type water cooler/heater was applied, the rate of change was the same with both cooling and heating. In all four of the scenarios, the heat source capacity turned out to decrease due to retrofit. In Scenario I and IV, the reductions were the biggest, 35.7% and 42.8% respectively, due to the strengthening of the legal requirements for skin heat insulation performance in 2015. Also, in the case of the second retrofit of Scenario I, there were no changes in the capacity of the heat source as compared to the first retrofit.

**Table 13.** Review of the capacity of absorption water heater/cooler for retrofit in each scenario.

| Scenario | | Cooling | Heating | With Retrofit Change Rate (%) |
|---|---|---|---|---|
| | | Equipment Capacity (kW) | Equipment Capacity (kW) | |
| **Scenario I** | **1985** | 4922 | 4430 | −35.7 |
| | **2000 \*** | 3164 | 2848 | |
| | **2015 \*** | 3164 | 2848 | - |
| **Scenario II** | **1990** | 4220 | 3790 | −16.6 |
| | **2005 \*** | 3516 | 3162 | |
| **Scenario III** | **1995** | 3938 | 3546 | −19.7 |
| | **2010 \*** | 3164 | 2848 | |
| **Scenario IV** | **2000** | 4922 | 4430 | −42.8 |
| | **2015 \*** | 2814 | 2534 | |

\* When retrofit was carried out.

## 4. Conclusions

The purpose of this study was to identify the changes in internal heat sources by time and to review the heat source capacity at the time of retrofit as compared to the initial design, as a preceding study for a study to explore options for retrofit with a minimized replacement of heat source equipment.

The study target was a large office building in Seoul Korea, and RTS-SAREK, which is widely employed, was applied to conduct the study. First, literature reviews were conducted to determine the operational status of internal loads by time. Through the review results, changes in the internal heat loads were identified quantitatively, indicating that the OA load increased by 49.6% over the last 30 years. While the lighting load was reduced by 47% due to LED light replacement. After this, the design criteria of envelope insulation performance were studied, and the insulation performance was improved by 53% for the outer walls, 69% for the roof, 50% for the floor, and 40% for the windows. The studied results of internal and skin loads were applied to RTS-SAREK to calculate the maximum cooling and heating loads.

Furthermore, the maximum cooling and heating loads were calculated for four retrofit scenarios considering the poor old building criteria and allowable retrofit times. The calculated results verified that Scenario IV, belonging to the period from 2000 to 2015 had the largest load reduction.

Also, the infiltration load decreased approximately 40~60% over the all scenarios, which had the most significant influence on the reduction of the maximum cooling and heating loads. Lastly, based on the calculated maximum cooling and heating load, the heat source capacity was calculated. In the case of Scenario IV, the capacity was reduced by 43% at the time of retrofit as compared to the time of initial design.

The study results showed that the maximum load varied according to the building completion time, which was due to the quantitative input data which was thoroughly studied and based on the building's operational status. Since this input data considered the operational status of the actual building, which was different from the design criteria, they can be applied as an index to determine an approximate load level of buildings aged less than 30 years. As described above, the maximum load of buildings varies according to design time and the elapsed period, which should be taken into consideration at the time of retrofit.

Due to the current trend, the design criteria for buildings are being strengthened each year, the maximum cooling and heating load seems to be bound to decrease. However, a simple replacement of the heat source would reduce the partial load rate of the equipment, resulting in an unnecessary use of energy. Therefore, a reconsideration of the skin and internal heat load and the heat source capacity at the time of retrofit is believed to be a contribution to the enhancement of the energy performance of the building. Based on the results of this study, the researcher plans to identify strategies for heat source systems with minimal equipment replacement.

**Author Contributions:** K.-s.P. and H.-y.K. calculated the data; Y.-h.S. conceived and designed the methodology the paper; H.K. found literature and wrote the paper.

**Funding:** This work was supported by the National Research Foundation of Korea (NRF) grant funded by the Korea government (MSIT) (No.2017R1A2B2006424).

**Conflicts of Interest:** The authors declare no conflict of interest. And the founding sponsors had no role in the design of the study; in the collection, analyses, or interpretation of data; in the writing of the manuscript, and in the decision to publish the results.

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
