# Peer review of "A Study on the Changes in the Heat Source Capacity and Air-Conditioning Load due to Retrofit; Focusing on a Large Office Building in Korea"

_energies, doi:10.3390/en12050835_

Round 1
Reviewer 1 Report
The paper is on a highly debated topic in the academic community. In general it lacks the level of originality and innovation to be published in an international scientific journal. In detail the major concerns are the following:
The paper lacks a proper review of the scientific literature in the field. The paper includes an introduction paragraph, where some references to local Korean studies are included.
The research question is not clearly defined, and it is required an higher level of detail regarding the originality of the research.
The methodology adopted is not clear. The authors refer to a simulation study carried out a local simulation code (RTS-SAREK) widely used in industry in Korea. The authors have not included any result of calibration of the model nor any information on the validation of the methodology and of the simulation tool.
Some of the technical terminology is not totally correct. For example table 5 includes the values of the "skin load heat insulation", which is in reality to be referred as "thermal transmittance of envelope components".
Author Response
The paper is on a highly debated topic in the academic community. In general, it lacks the level of originality and innovation to be published in an international scientific journal. In detail the major concerns are the following:
The paper lacks a proper review of the scientific literature in the field. The paper includes an introduction paragraph, where some references to local Korean studies are included.
à In this paper, previous studies in nations other than Korea were added. Since this study was conducted based on the design criteria and office buildings in Korea, those previous studies were added as a basis to support this paper.
The research question is not clearly defined, and it is required a higher level of detail regarding the originality of the research.
à The detained contents about the research questions and needs were added and the purpose and originality of this study were added to improve the paper.
The methodology adopted is not clear. The authors refer to a simulation study carried out a local simulation code (RTS-SAREK) widely used in industry in Korea. The authors have not included any result of calibration of the model nor any information on the validation of the methodology and of the simulation tool.
à The details about RTS-SAREK applied in this study were added. In addition, the application scope in details, a range of possible calculation range of the program, and the calculation process was described for the understanding of the program for readers. Moreover, the active use and validation in the study areas were presented by adding research results and the current status of the actual design cases.
Some of the technical terminology is not totally correct. For example, table 5 includes the values of the "skin load heat insulation", which is in reality to be referred as "thermal transmittance of envelope components".
à The technical terminology was corrected accordingly.

Reviewer 2 Report
Researchers in this paper employed RTS-SAREK software which is not widely known and seems to be mainly used in South Korea. For the paper to be more suitable for the Energies journal, it is important first to establish confidence in the software by comparing its performance against another software that is well known in the field (world-wide) such as EnergyPlus.
To make the paper more useful to a wider audience, proper architectural drawings should be added to illustrate clearly the investigated floor plan showing any openings. It would also be helpful to present any photos of the building to show solid and glazed materials used and surface textures. Authors should also add 2D section architectural details showing building materials in roof, floor, walls, with dimensions.
Results presented in Figure 5 and Figurev6 are difficult to read/compare. They should be reproduced/presented in column/bar chart format (side-by-side) rather than stacked on top of each other.
Presenting important data/results in Tables, is not very helpful to readers. Most tabulated data/results can be better presented/plotted in a graphical format to make them more readable and comparable.
Conclusions don’t seem to reflect fairly the research findings. They could be rewritten/extended to include more factual data/results, with discussion.
Author Response
Researchers in this paper employed RTS-SAREK software which is not widely known and seems to be mainly used in South Korea. For the paper to be more suitable for the Energies journal, it is important first to establish confidence in the software by comparing its performance against another software that is well known in the field (world-wide) such as EnergyPlus.
à First, a case [Lee Seong Goo] that was added to improve the reliability of the program. In the revised paper, the following content was added showing that the program has been widely used in practice when designing a thermal source system in Korea. Since this study targeted office buildings in Korea, the application of RTS-SAREK used in actual thermal source system designs would represent the current status in Korea. Thus, the results of the program are applied.
To make the paper more useful to a wider audience, proper architectural drawings should be added to illustrate clearly the investigated floor plan showing any openings. It would also be helpful to present any photos of the building to show solid and glazed materials used and surface textures. Authors should also add 2D section architectural details showing building materials in roof, floor, walls, with dimensions.
à In the revised paper, the window and wall thickness were notated in detail in the floor plan in Figure 3. Since thermal transmittance, which is the legal design standard used in the outer wall and roof, was applied in this study, the detailed description of the materials was not required. This study assumed that the design and construction were conducted according to thermal transmittance that was suitable to the standard without the limitation of the material’s physical properties. Accordingly, this study determined that the selection of thermal transmittance for each of the materials that made up the outer wall and roof was not necessary. Thus, thermal transmittance of the entire outer wall and roof were applied.
Results presented in Figure 5 and Figurev6 are difficult to read/compare. They should be reproduced/presented in column/bar chart format (side-by-side) rather than stacked on top of each other.
à The detailed comparison by load item was added in the form of a graph regarding Scenario 1 in Figure 5. After this, the scale in the comparison graph of the overall maximum cooling and heating loads were made to be equal to raise the readability. Since the results of Scenarios 2 to 4 were relatively similar to that of Scenario 1, the values at the times of design and retrofitting were added as the label in the figure.
Presenting important data/results in Tables, is not very helpful to readers. Most tabulated data/results can be better presented/plotted in a graphical format to make them more readable and comparable.
à Tables 6 to 11 present input data results which are applied to simulations after quantifying the literature review results. Moreover, the detailed comparison on the literature review results was already done in a previous study [Kim Hye Mi], this study mentioned this in the paper and the comparison analysis target in the research results was limited only to the maximum load. In addition, a graph was added to present the results in Table 12 for readers to better understand the content.
Conclusions don’t seem to reflect fairly the research findings. They could be rewritten/extended to include more factual data/results, with discussion.
à The conclusions of this study were extended and re-written. The derived results and analysis findings were added.

Reviewer 3 Report
The authors should improve some minor issues prior to the acceptation on the journal.
The abstract must be improved, the goal of the paper is clear, but the expression must be amended.
It is needed to describe the RTS-SAREK software, at least describe in detail the nature of the software and the purpose and method used.
A complete discussion regarding why those scenarios are selected must be done. Also if the nature of the modeling process done in RTS-SAREK is stochastic, an statistical analysis must be performed, otherwise must be justified the nature of the results in each scenario.
The title should mention that this study focuses on Korea, this would make easier for a foreigner to understand what it's exposed and the climate zone used. The loads showed on figure 6 should be exposed using the same scale, this would make easier to compare them (line 327).
The English of the paper can be improved, as example, on line 56 and on line 58 the two sentences start with therefore, the second one should be changed.
Author Response
The authors should improve some minor issues prior to the acceptation on the journal.
The abstract must be improved, the goal of the paper is clear, but the expression must be amended.
à The abstract was modified to improve the expression.
It is needed to describe the RTS-SAREK software, at least describe in detail the nature of the software and the purpose and method used.
à The characteristics of the RTS-SAREK software and the purpose of the use were described in Study Method and scope in Chapter 2, and the input values applied to the program are presented in Tables 1 and 3.
A complete discussion regarding why those scenarios are selected must be done. Also, if the nature of the modeling process done in RTS-SAREK is stochastic, a statistical analysis must be performed, otherwise must be justified the nature of the results in each scenario.
à The selection process of the scenarios was described in detail, and the reason for the selection of four scenarios was also described. In addition, a detailed analysis was conducted with regard to Scenario 1 and the graph was added to justify the study results of that scenario. Moreover, a comparative analysis was conducted with regard to the differences between the scenarios, thereby adding comparative analysis results.
The title should mention that this study focuses on Korea, this would make easier for a foreigner to understand what it's exposed and the climate zone used.
à Focusing on Korea was added to the title of this paper, and the explanation about climate zone in Korea was also added to the study scope.
The loads showed on figure 6 should be exposed using the same scale, this would make easier to compare them (line 327).
à The scales in the graphs for cooling and heating in Figure 6 were made equal to enable the comparisons to be made easily and quantitatively.
The English of the paper can be improved, as example, on line 56 and on line 58 the two sentences start with therefore, the second one should be changed.
à The English expressions were modified by revising the introduction.

Round 2
Reviewer 1 Report
The paper has been improved significantly and has now reached a sufficient level to be published.
Author Response
We have checked again all typographical errors, tables, figures, etc. in the manuscript

Reviewer 3 Report
The paper has been improved and now is easier to understand. in the second line of the abstract is not clear what it wanna express "this makes a review of the capacity a necessity previous to the heat source system being replaced", this sentence could be canceled or improved. The rest is clear.
Line 86 is repeated to closely "off".
Between line 99 to 101 is repeated to closely 15 years.
Line 133 what refers to number 9.
Line 146 optimum maybe should be optimal.
Line 176 table 1 title should be under the table.
Now is much clearer the area of study, better, this has been improved correctly.
Figure 6, letter c is in a different scale from the rest.
Author Response
Line 86 is repeated to closely "off".
è We have deleted redundant parts of sentences and simplified them for easy reading
Between line 99 to 101 is repeated to closely 15 years.
è 15 years for the retrofit permission period and the heat source service life are separate, therefore we have replaced sentence for easy understanding.
Line 133 what refers to number 9.
è We have summarized the Reference 9 and described in the main manuscript.
Line 146 optimum maybe should be optimal.
è We have replaced with appropriate word as reviewer’s comment.
Line 176 table 1 title should be under the table.
è We have checked again Journal drafting format about all figures and tables etc.
Now is much clearer the area of study, better, this has been improved correctly.
Figure 6, letter c is in a different scale from the rest.
è We modified the scale of Figure 10 (c) to be the same as the others.
